# Innovation Process for Optical Face Scanner Used to Customize 3D Printed Spectacles

**DOI:** 10.3390/ma15103496

**Published:** 2022-05-13

**Authors:** Cristian Gabriel Alionte, Liviu Marian Ungureanu, Tudor Mihai Alexandru

**Affiliations:** 1Mechatronics and Precision Mechanics Department, Faculty of Mechanical Engineering and Mechatronics, University Politehnica of Bucharest, 060042 Bucharest, Romania; cristian.alionte@upb.ro (C.G.A.); tudor.mihai2098@yahoo.ro (T.M.A.); 2Mechanisms and Robots Theory Department, Faculty of Industrial Engineering and Robotics, University Politehnica of Bucharest, 060042 Bucharest, Romania

**Keywords:** 3D printing, optical scanning measurement, 3D spectacle frames

## Abstract

Many people for different reasons end up wearing glasses to correct their vision. From time immemorial, there has been an unquestionable ability to associate people with glasses. Designing the glasses according to the physiognomy of each person opens a new path for a completely new optical experience. The frames are designed to fit perfectly on the face, are comfortable on the nose, and are positioned at an optimal distance from the cheeks and eyelashes. Three-dimensional printing technology offers the possibility to customize any form of glasses at a low cost with average quality. In this type of technology, the printer receives a digitized model of the spectacle frame (usually in STL file format) that must meet the parameters related to the wearer’s anatomy. Therefore, this paper presents an innovative process, an optical method used to scan the wearer’s face to design a parameterized design of the spectacle frames. The procedure has a measurement phase for quantifying the anatomical features of the wearer’s face, a para-metric design phase of the glasses for adjusting the design parameters according to the anatomical characteristics, and a manufacturing phase in which the custom eyeglass frame will be manufactured using 3D printing technology. The aim of this study was to create an innovative process that could be tested as an educational 3D printing system that could be used by undergraduate students (studying under an optometry program), a process that would begin at optometric prescription stage and can be used in the educational laboratory of the Department of Mechatronics and Precision Mechanics from the Politehnica University of Bucharest. Using this method we obtained a custom spectacle frame that can be prototyped using 3D printing. The 3D-printed polylactic acid (PLA) frames are lightweight, flexible, durable, and the innovative photogrammetry process gives designers the ability to create custom designs that cannot be created with traditional manufacturing techniques.

## 1. Introduction

Spectacle frames are easy to manufacture using 3D printing methods because this technology permits their manufacturing in many designs, sizes, and shapes [1]. The shape parameters are linked with the anatomical features of the wearer for optimal positioning of the lenses in front of the eyes to align the center of the lens with the optical eye axis and to achieve improved monocular and binocular vision according to the prescription of the optometrist. Therefore, the spectacle design must respect certain dimensions, be comfortable around the temples and on the nose, must be aesthetic, and must fit the parameters of the anatomical features that allow the spectacles to be worn effortlessly. Each person has a different facial anatomy and the 3D printing of spectacle frames seems to be a good solution that solves this problem through easy customization without having to make compromises [2]. The concept, design, and prototyping are not only accurate, but also allow the realization of unique designs and the spectacle frames can be manufactured in many sizes, shapes, and colors. These characteristics can be selected and virtually verified according to the customer’s preferences and anatomy.

Although 3D printing technology for spectacles frames is relatively new, large companies have not hesitated and have seen the great potential it has in developing unique products, thus several competitors have appeared on the market [3,4] to produce 3D-printed eyeglass spectacles frames.

Yuniku [3] used parametric design automation and used photographs of the wearer to build a 3D digital anatomy. The model was used to design the spectacles frame. In this way, advanced software (designed by Hoya) uses facial and visual data to determine the ideal position of the lens relative to the eyes. From the software, the obtained parameters are sent to the 3D frame design software, which adapts the frame around the lens, depending on the unique characteristics of the wearer’s facial features. The design of the frame, the color, and the finishing can be adjusted to suit the individual style of the client, guided by the expertise of the eye care specialist. The integrated software solutions work in the background to ensure that both the ideal positioning of the lenses and the fit of the frame are preserved. The digital model of the spectacles frame is sent to a 3D printing system. This system has the disadvantage of using wearer photos and the 3D digital anatomy is not very accurate.

Monoqool [4] is another 3D printing company from Copenhagen. With their 3D-printed spectacles frames they bring the elegant and functional elements of Danish design to the field of eyewear production. The company has worked with 3D printing since 2008, initially using technology to prototype new models of glasses before moving to large-scale production. Like other brands of 3D-printed glasses, Monoqool relies on selective laser sintering (SLS), which is a type of powder bed fusion (PBF) [5], to produce its spectacles frames. Their goal is to obtain high-quality spectacles frames utilizing a new mechanism to create unique frames without hinges. Monoqool differs from its competition by the design of the frames of their spectacles, which are very thin and light and use a screwless hinges patented mechanism. Furthermore, the material is recyclable, using 85% surgical stainless steel to produce the arms and nasal saddle, and they reuse 98% of the powder from which the spectacles frames are printed [4].

YourEyewear [6] is a concept developed by More Eyewear, the producer of several trademarks: Vingino, State of Art, Capo, and Piet Hein Eek Eyewearce. They work exclusively with entrepreneurial optometrists because they consider the profession of the optometrist to be a valuable profession for people who wear glasses. The spectacles frames are made of high-quality nylon. The characteristics of the spectacle’s frames are given by the uniqueness of each pair of glasses, the customer can choose any color or color combination. Each frame can be finished with a texture such as leather or precious stones. To meet the most drastic wishes of customers, each pair of glasses also has its own serial number to provide even more value and allow the option to add the initials, name, or favorite saying on the arms of the glass’s spectacles frames [6].

## 2. The Process of Manufacturing Spectacles Frames Using 3D Printing

Three-dimensional printing technology can be applied in the optometric field following several phases. In the first phase, an ophthalmological examination must be performed by an optometrist to achieve a lens prescription. The lenses can be monofocal or multifocal (bifocals, trifocals progressive, or degressive) [7] according to the wearer’s conditions or needs. The wearer’s conditions are linked with the type of eye seeing problems:Myopia—seeing problems only at a far distance [8]hyperopia—seeing problems only at a close distance [9]astigmatism—blurry or distorted vision distance [10]presbyopia—seeing problems only at far and close distances due to accommodation reflex problems [11].

The eye seeing problems are important because the position of the lens center must be on the anatomical optical axis of the eye [12].

After the problems and needs of the wearer are identified, the face of the client is scanned and their unique facial features are identified. Using 3D specialized software, a virtual model of the face is built. On the virtual model, a 3D model of the frame of the spectacles is made. In this phase, the virtual fitting and placement of the lenses in the ideal position are undertaken, depending on the optical parameters, facial features and the type of the spectacles. Also, the aesthetic customization settings are performed.

In the final phase (the manufacturing), the frame is exported in STL file format that 3D printers can use [13]. The model is divided into layers/slices in the processing software and transformed into G-code used by a 3D printer in the manufacturing process.

The most used technology is selective laser sintering (SLS), but selective laser melting (SLM) printers can also be used [14]. SLS technology uses a high-power laser beam to melt powders in successive layers, thus obtaining the desired 3D model. Based on the 3D model data, the mobile laser beam synthesizes the powder layers on the construction platform inside the 3D printer tank. After finalizing the first section, the platform on which the 3D model is built is lowered inside a tank so that the next section can be made, then the process is repeated until the realization of the 3D model, according to the STL file.

During printing, the 3D model is permanently framed in powder, which allows the printing of extremely complex shapes without requiring the printing of media, as in the case of 3D printers with technology modeling by extrusion [15]. The remaining powder in the construction tank can be reused in subsequent prints. The main disadvantage of SLS is that the obtained model is porous and requires further finishing to make it more rigid [16]. The accuracy of the printed parts is very good, the finishing of the printed surfaces is superior, the printing speed is high, and the spectacles frames can be subjected to post-production treatment in several stages to obtain colors and finishing. SLS technology allows the use of a wide range of materials, such as nylon, polyamide, and other composites [17].

SLM technology is a sub-branch of SLS technology with a similar manufacturing process and uses metal powders from a harder material such as steel, titanium, and other alloys, which are melted and welded together using a high-power laser. The thin layers of metal powder are successively melted and solidified microscopically inside a closed construction chamber containing inert gas (nitrogen or argon). Upon completion, the 3D model is removed from the construction chamber and subjected to finishing treatment [14].

## 3. Materials and Methods

### 3.1. 3D Optical Scanning System

Before the presentation of the actual design, there follows a review of the elements that underly the operation of this device, such as: photo camera (DSLR), camera lens, materials used in the parts of the scanning device, the motor that makes the movement of the whole system possible (in our case a stepper motor), and the electronics used to control the system (Arduino development board).

To make it possible to scan the face of the wearer a DSLR camera type Canon EOS 80D was used [18]. The camera can acquire up to 7 photos per second in a low-light environment. The autofocus has 45-point, of which 27 are cross-type, thus ensuring clear images regardless of lighting conditions or moving subjects. This focusing system covers a large part of the frame while also being much more sensitive. The device has a CMOS sensor (APS-C format) with a resolution of 24.2 megapixels and a brightness ranging from ISO 100 to ISO 16000. These characteristics allow for extraordinary images that are rich in detail even in poor lighting conditions, an aspect that favors the topography of a human face.

Tokina AT-X 116 PRO DX [19] is a professional lens with a shutter angle of 104°–82° and a def/2.8 aperture for a good resolution in low light. Based on the optical design of the AT-X 124 PRO DX (12–24 mm f/4), the new AT-X 116 PRO DX has a slightly shorter zoom range to maintain optical quality at open apertures.

The design of the device used for scanning the human face began with the idea of a curved slider. A semi-hard copper pipe of 5 m × 22 mm × 1 mm was used for the guide rail for the trolley that supports the optical system (Figure 1). Copper pipes are safe, durable, and reliable due to their excellent mechanical and physical properties. The copper withstands extreme temperatures and pressures and is an easy material to handle and bend. It also has a low coefficient of thermal expansion, high corrosion resistance, does not lose its properties over time, and is the only plant material that, due to its nature, is antibacterial and hygienic.

The size of the slider guide had to cover a little over half the diameter of the circle so that the camera could perform the full scanning of the bust during the image acquisition process.

The whole system was sustained by a rigid support made from wood (OSB board), with a thickness of 8 mm (Figure 2).

To be able to fasten the rail (copper pipe) at an optimal distance, a set of 7 spacers was used, evenly distributed along the entire length of the rail. The spacers were manufactured from duraluminium rods 130 mm long and 8mm in diameter with M4 threads at both ends so that they could be easily mounted (Figure 3).

For the design of the trolley that would support the optical camera, the software SketchUp Pro was used [20]. For the correct assessment of the shape and dimensions of the trolley, it was important to diminish the torque on the copper pipe as much as possible. Thus, the final product had the dimensions 250 mm × 140 mm × 70 mm. The digital design and the final product are shown In Figure 4.

Three wheels were manufactured for the sliding and traction system of the trolley. The wheels that were to slide on the copper pipe and their design had to cancel the moment of rotation of the trolley. The trolley had a motor support that would serve as its coupling and decoupling system. For the wheels, a material was manufactured that had good mechanical properties and was very easy to manufacture. This material consists of multiple layers of cotton, glued together with a polymerizing resin (in our case epoxy resin). The wheels were designed in SketchUp Pro and used SKF 608 ZZ bearing [21]. Due to the small dimensions of the trolley, we created a system that eliminated the moment of rotation on the copper pipe. This was made from two aluminum supports in an L shape which was mounted in front of the two wheels. Inside the support was melded a system of bearings mounted tangentially on the surface of the copper pipe. These supports were in contact with the copper pipe due to the help of three elastic elements (Figure 5).

To move the whole assembly, it was necessary to introduce a compact, light, powerful and precise motor. Because stepper motors are generally used in such applications where precise position control is desired, we used one (HANPOSE 17HS4401) [22] with a size of 40 mm × 40 mm × 40 mm, an accuracy of 1.8 degrees/step (200 steps/rotation), and a weight of only 280 g to exert a torque of 40 N × cm.

Figure 6 shows the electrical diagram of the motor-driven assembly of the carriage supporting the camera. Using the button (6) to start the optical scanning process, the Arduino (2) receives a digital signal, and the red led is activated (4). A signal is sent to the driver A4988 (3) to move the engine (1) to the end of the trajectory. After each step of the engine, the camera receives a signal from the Arduino to take a picture. When the trolley reaches the limit switch (5), the system turns off and the device moves to the reference start position. To always have the same starting point, the system performs a calibration in which it moves a few mm to the opposite side of the limiter, then slowly take a step until the stroller touches the limiter and receives a digital signal.

### 3.2. A 3D-Printed Spectacles Educational System

This paper, according to the conditions defined up to this point, will present a method and a prototype for scanning the face of the spectacles wearer and the creation of a 3D printed frame model will be exemplified.

Another purpose of this study was to establish the feasibility of a new method of manufacturing custom glasses, for those who want to create unique spectacles frames and for people with facial malformations, using 3D printing technology.

To make the prototype of the face topography, it was necessary to design a device that allows the taking of a multitude of photos in a 180° range, using a photo camera (DSLR type) and, after the scanning is completed, for the obtained images to be transmitted in specialized software to a 3D digital head that is created and on which the spectacles frames can be adapted.

In the design phase, the following aspects must be considered:the whole assembly must be modular to be easy to transport and assemble.all modules must be optimized to use as little space as possible.the actuation control module will be provided with a removable connector to facilitate the modularity of the assembly.easy-to-purchase materials must be used to reduce the production cost of the educational prototype.

The printer used for the spectacles frame was a Tevo 3D printer [23]. The fused deposition modeling technique was made for the parts manufactured in plastic through the Ultimaker Cura^®^ computer program that allows generating the G code for the 3D printing machine [24]. After importing the STL format model, we proceeded to configure the parameters of in Cura^®^ (Figure 7).

The position of the spectacles frame on the building platform is presented in Figure 8.

### 3.3. Optical Face Scanner Method Used to Customize 3D Printed Spectacles

Using advanced 3D printing creates the possibility to customize the spectacles frames, starting from the optimal optical face scanner that evaluates facial features. Mashroom and MeshLab software calibrate the optimal lens position for the best visual performance. The spectacles frames are automatically tailored to fit perfectly to an individual’s facial features.

To provide a better understanding of the process of making 3D spectacles frames, a diagram is presented in Figure 9, intended to reflect a global vision of the whole process.

## 4. Results

In the process of optical scanning of the human face, the camera and the mentioned lens were used. After mounting the optical camera on the built carriage, several tests were performed to determine the proper number of photos needed for a complete 3D model. After several attempts, it was concluded that 84 photos were enough to compose a good quality 3D image. For the environmental conditions under which the experiments were conducted, the following camera settings were used: 16mm focal length, ISO-100, F2.8 aperture, 1/125 shutter speed (fast enough to cancel any movement caused by starting and stopping the system).

For the image processing process, the open-source reconstruction software 3D Meshroom offered by AliceVision [25] was used. The 3D model of the main object can be built from a set of images taken from different angles. This software practically analyzes the images, using some photogrammetric algorithms, calculates the surface points, and automatically produces a 3D model (Figure 10). The model can be exported as a file in OBJ file format. The software has a 3D viewer integrated, where the final model can be seen. Figure 11 shows the 3D model generated from all photos.

The 3D model can be rotated and analyzed using the mouse and the model was saved into an OBJ file format that could be further processed.

This software records all the operations that can be seen in the command window. To be able to go further, the OBJ file will be automatically transformed as a sum of textures into a solid object type (STL file format) using the MeshLab program [26]. MeshLab software is an open-source program, developed by the ISTI-CNR research center. It can be used for modelling, texturing, reconstructing, and adjusting 3D objects, which can generate and export STL files which are sent to the 3D printer.

To adjust the 3D model from the 3D image processing to correspond to reality, the interpupillary distance of the wearer of the spectacles was measured [27], and the 3D rendering was scaled so that the virtual one corresponded to reality (Figure 12). It should be noted that the interpupillary distance is not normally the same for both eyes so the frame of the spectacles should be asymmetric.

After the spectacle frame has been adapted in the 3D design software for the user’s face, based mainly on the interpupillary distance, it is tested on the client’s 3D bust (Figure 13).

Based on the 3D model, a series of spectacles frames have been designed that can be subsequently adjusted to the needs of each client, using the assisted design software. The spectacle frame consists of a lens holder, called the front, the arms for fixing the glasses behind the ear, and the nasal saddle. In the design process, the spectacles frames should be aesthetic, light, resistant to physical and chemical agents, non-flammable, and durable. For the correct assessment of the parameters related to shape and dimensions of the spectacles frames [28], the Boxing system was used, which is mainly used in the USA, Germany, as well as in Romania. After finishing the reconstruction process, the 3D model in Figure 14 was obtained.

Finally, after sending the Gcode file to the 3D printer using the parameters for a Tevo 3D printer [23] we obtained a 3D-printed spectacles frame (Figure 15).

## 5. Conclusions

In the paper, we have presented an innovative process for an optical face scanner used to customize 3D printed spectacles, a low-cost method and system based on 3D printing technology for educational purposes and people with facial malformations.

To make the educational system it was necessary to design and build a device that allows a multitude of photos to be taken on a 180° range, using a DSLR camera (Canon EOS 80D DSLR) with a wide-angle optical objective (Tokina AT-X 116 PRO DX) and to process the obtained images in specialized software (SketchUp Pro) to create a 3d bust of the wearer on which the spectacle frames will be adapted. After several attempts, it was concluded that 84 photos were enough to compose a good quality 3D image, as fewer photos are unable to create the prototype and more photos slow down the entire process. Also, the best results of the photos were obtained using the following camera settings: 16mm focal length, ISO-100, F2.8 aperture, 1/125 shutter speed, faster enough to cancel any movement caused by starting and stopping the system.

The system must be modular, easy to transport, small in size, and made with easy-to-purchase materials to reduce the production cost. The optical system is moved on a guide rail using a mobile carriage. The movement is generated by a HANPOSE 17HS4401 stepper motor controlled by an Arduino controller and A4988 driver. The image processing was performed using the Meshroom software and building a 3D model of the main object from a set of images taken from different angles.

For the correct assessment of the shape and dimensions of the spectacle frames, the Boxing system can be used. The user’s interpupillary distance was measured to scale the 3D model so that the virtual model corresponds to reality. After the spectacle frame has been adapted using a 3D design software and the spectacles virtual model is sent to the 3D printer, the spectacles frames are obtained (Figure 15). Also, a catalog with finishing levels and colors can be created to complete the advantages of the material and technology.

The following issues were identified as possible limitations of this study: an appropriate 3D printing equipment and a DSLR camera are needed, and the optimum number of photos taken, as less than 84 cannot generate the prototype and more than that will slow down the process.

Designing the glasses according to the physiognomy of each person opens a new path for a new optical experience. As the frames are designed to fit perfectly on the face, even for those persons with facial malformations, 3D printing technology offers the possibility to customize any form of glasses at a low cost with average quality. Therefore, this paper presented an innovative process for an optical method used to scan the wearer’s face to design a parameterized design of the spectacle frames.

## Figures and Tables

**Figure 1 materials-15-03496-f001:**
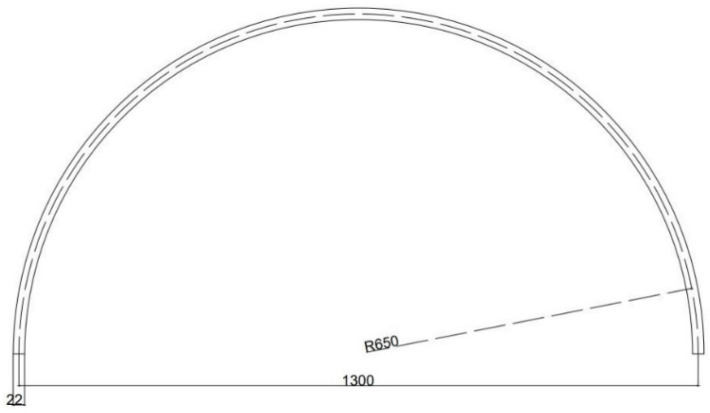
Design of the semi-hard copper pipe of 5 m × 22 mm × 1 mm.

**Figure 2 materials-15-03496-f002:**
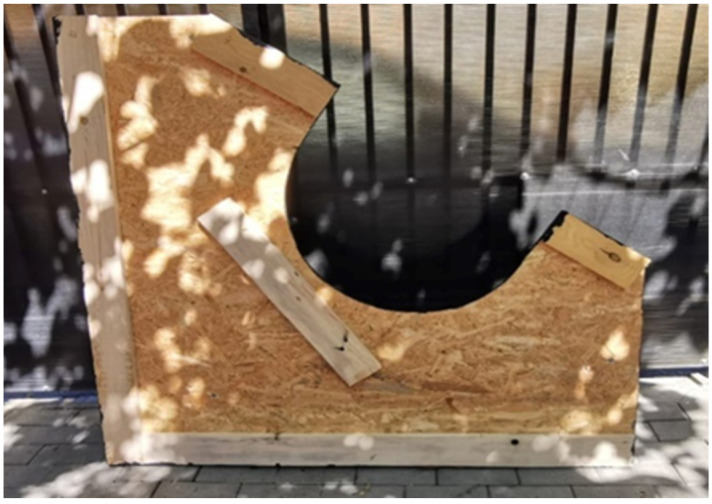
Rigid support manufactured from OSB board.

**Figure 3 materials-15-03496-f003:**
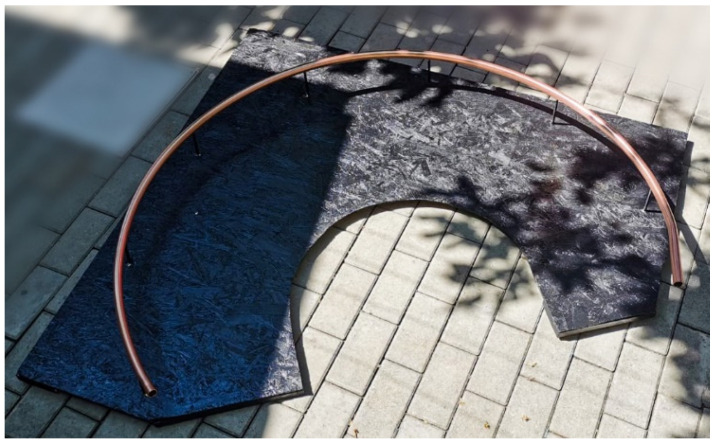
The rail fastened on the support board.

**Figure 4 materials-15-03496-f004:**
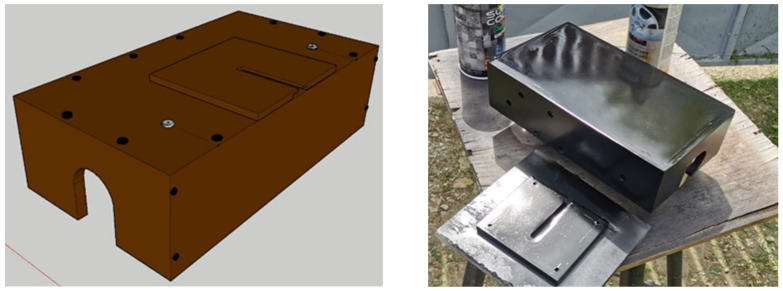
The trolley design in SketchUp Pro and the final prototype.

**Figure 5 materials-15-03496-f005:**
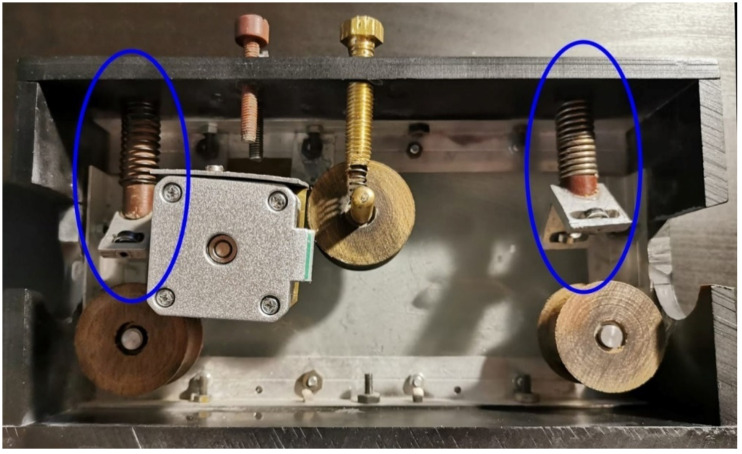
Supports to eliminate the torque.

**Figure 6 materials-15-03496-f006:**
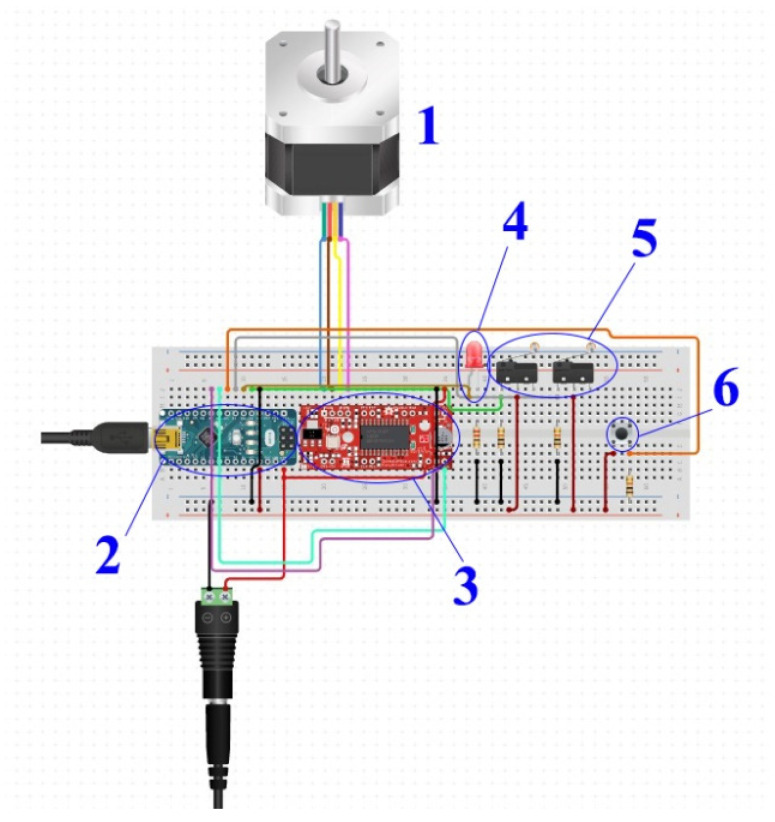
Electrical diagram: 1—17HS4401 motor, 2—Arduino Nano, 3—a4988 driver, 4—LED, 5—switch, 6—Button.

**Figure 7 materials-15-03496-f007:**
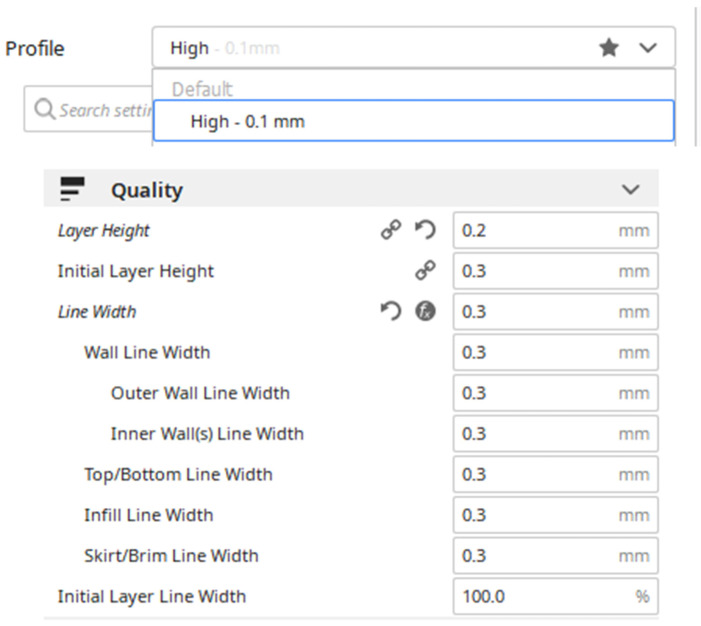
Configuration parameters in the Ultimaker Cura^®^ software for manufacturing the frame of the spectacles using polylactic acid (PLA) as the material.

**Figure 8 materials-15-03496-f008:**
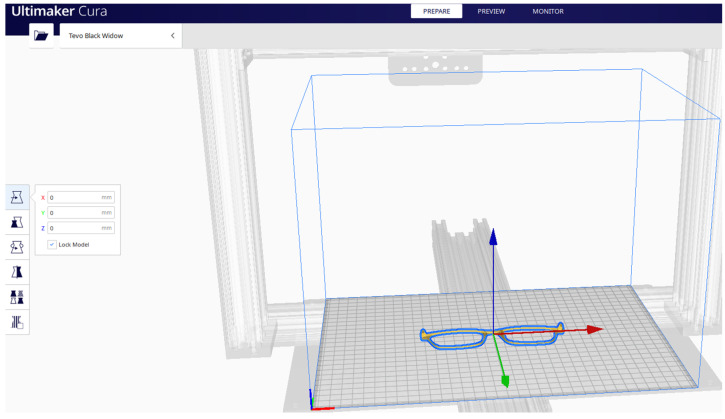
Spectacles frame parameters in the Ultimaker Cura^®^.

**Figure 9 materials-15-03496-f009:**
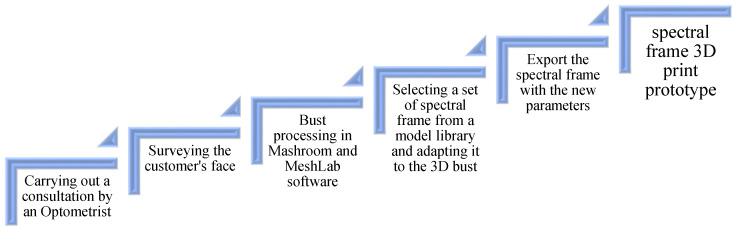
Schematic of the process of making 3D spectral spectacles frames.

**Figure 10 materials-15-03496-f010:**
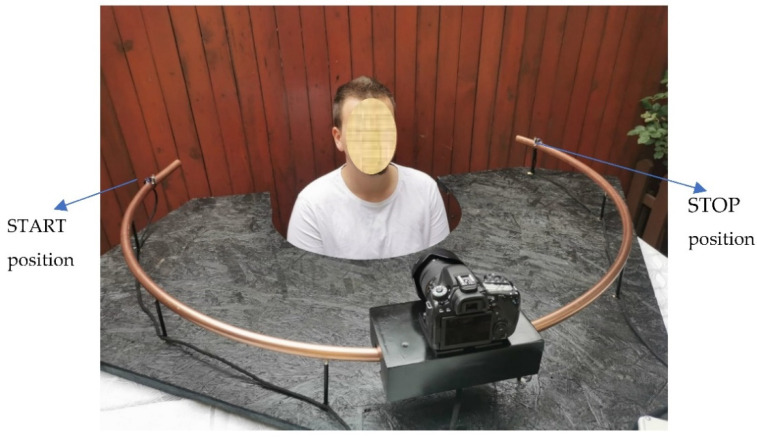
Process of human face scanning—between START and STOP positions.

**Figure 11 materials-15-03496-f011:**
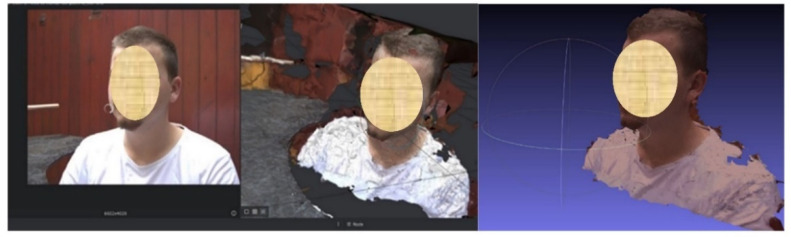
The final 3D model ready to be able to test the spectacle frame.

**Figure 12 materials-15-03496-f012:**
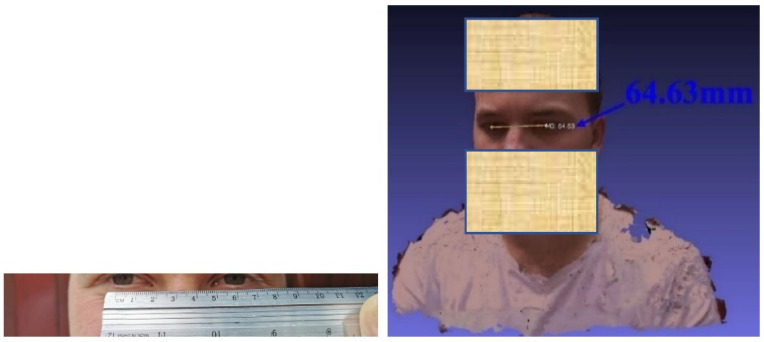
Measuring the client’s pupillary distance.

**Figure 13 materials-15-03496-f013:**
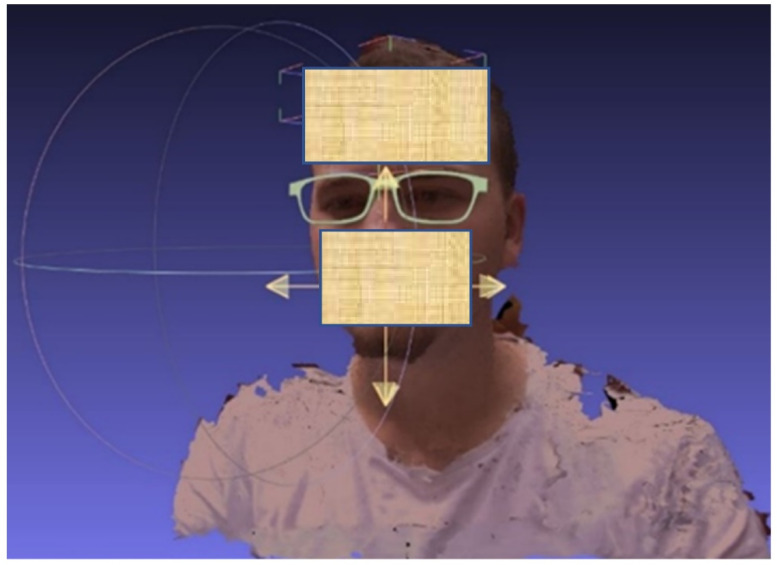
Final frame of the spectacles adapted to the 3D anatomical features of the wearer of the spectacles.

**Figure 14 materials-15-03496-f014:**
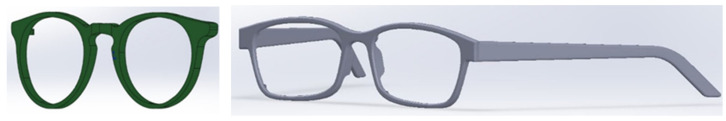
Design 3D models from spectacles frames.

**Figure 15 materials-15-03496-f015:**
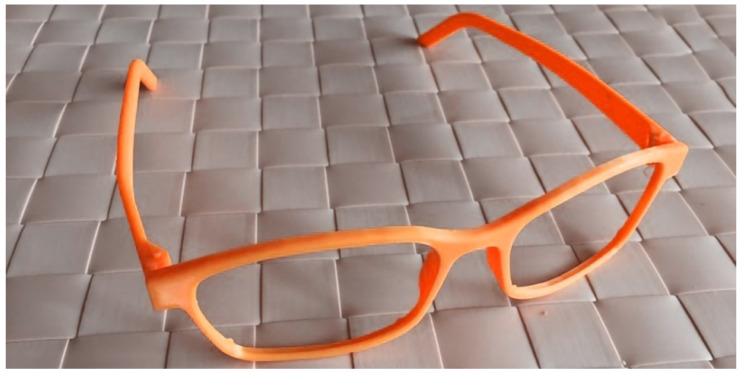
Result—3D-printed spectacles frames.

## Data Availability

Not reported yet.

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
