# Peer review of "Innovation Process for Optical Face Scanner Used to Customize 3D Printed Spectacles"

_materials, 2022, doi:10.3390/ma15103496_

Round 1

Reviewer 1 Report

The manuscript reports a process and equipment for optical face scanning to be used for the customization of spectacles by 3D printing (fused filament fabrication). The research topic is of interest and importance for optometry and additive manufacturing; however, the work described in the paper must be revised to bring more clarity and address a few gaps. The following recommendations need to be considered.

  1. Abstract: The abstract is relatively weak and has some unnecessary details. It should precisely specify the research problem, motivation, objectives, accomplishments, and future implications. The abstract needs revision.
  2. Page #8 (line 256-258): Please elaborate on the “image processing” and “STL file developments” steps; it is a bit unclear and needs a better explanation.
  3. The authors mentioned the “development of a parameterized design of the spectacle frames” as one of the objectives. It has not been presented/ explained.
  4. Please provide more details on the 3D printing of the spectacles using FDM, e.g., the material used, sliced model/ toolpath, part orientation, printed part quality, etc. This detail should be included in the "Materials and Methods" section.
  5. The quality of writing needs to be considerably improved throughout the paper to bring clarity and better flow.

Reviewer 2 Report

This paper presents a method of scanning by photogrammetry the face of a wearer of the spectacles and the creation of the spectacle frame model.

As general remark, this paper do not cover the scope of Materials Journal, as research related to materials. Also, the material characterization techniques are missing.

From my point of view there are many aspects to improve:

1. Abstract does not contain any quantitative data about the results. Abstract should be revised according to the template. (1) Background: Place the question addressed in a broad context and highlights the purpose of the study; (2) Methods: Describe briefly the main methods or treatments applied; (3) Results: Summarize the article's main findings; and (4) Conclusions: Indicate the main conclusions or interpretations.

2. Some methods are presented in the Results section, as “The fused deposition modeling technique is made for the parts manufactured in plastic through…”

The structure of the paper should be revised taken into accord the template. Thus, Materials and methods, and Results should be distinct sections. The Conclusions should be written based on the Results.

3. The first three paragraphs from Materials and Methods are literature and should be moved to the Introduction

4. A 3D printing machine is mentioned in the paper but details about it are missing. Also, the materials used to print the spectacle frame are missing.

5. An example of 3D printed spectacle frame is shown in Figure 14, but details about how it was manufactures are missing. The pre-processing stage within Ultimaker Cura software should be in detail presented. How were positioned the frame on the build platform?

6. The conclusions section should be carefully written based on the results.

7. What is new in this research? Please mention it in the conclusion section.

8. Are the limitations of this study noted? The limitations of this study should be discussed.

Round 2

Reviewer 2 Report

All the comments are addressed well and utilized to improve the manuscript. The manuscript is acceptable.